# Reflections on COVID-19: A Literature Review of SARS-CoV-2 Testing

**DOI:** 10.3390/vaccines13010009

**Published:** 2024-12-26

**Authors:** Chin Shern Lau, Helen M. L. Oh, Tar Choon Aw

**Affiliations:** 1Department of Laboratory Medicine, Changi General Hospital, 2 Simei Street 3, Singapore 529889, Singapore; 2Department of Infectious Diseases, Changi General Hospital, 2 Simei Street 3, Singapore 529889, Singapore; 3Department of Medicine, National University of Singapore, Singapore 117599, Singapore; 4Academic Pathology Program, Duke-NUS Graduate Medical School, Singapore 169857, Singapore

**Keywords:** SARS-CoV-2, COVID-19, immunoassays, antibodies, vaccination

## Abstract

Although the Coronavirus disease 2019 (COVID-19) pandemic has ended, there are still many important lessons we can learn, as the pandemic profoundly affected every area of laboratory practice. During the pandemic, extensive changes to laboratory staffing had to be implemented, as many healthcare institutions required regular screening of all healthcare staff. Several studies examined the effectiveness of different screening regimens and concluded that repeated testing, even with lower sensitivity tests, could rival the performance of gold-standard RT-PCR testing in the detection of new cases. Many assay evaluations were performed both in the earlier and later periods of the pandemic. They included both nucleocapsid/spike antibodies and automated antigen assays. Early in the pandemic, it was generally agreed that the initial nucleocapsid antibody assays had poor sensitivity when used before 14 days of disease onset, with total or IgG antibodies being preferred over the use of IgM. Spike antibody assays gradually replaced nucleocapsid antibody assays, as most people were vaccinated. Spike antibodies tracked the rise in antibodies after vaccination with mRNA vaccines and became invaluable in the assessment of vaccine response. Studies demonstrated robust antibody secretion with each vaccine dose and could last for several months post-vaccination. When antigen testing was introduced, they became effective tools to identify affected patients when used serially or in an orthogonal fashion with RT-PCR testing. Despite the numerous findings during the pandemic period, research in COVID-19 has slowed. To this day it is difficult to identify a true neutralizing antibody test for the virus. An appropriate antibody level that would confer protective immunity against the plethora of new variants remains elusive. We hope that a summary of events during the pandemic could provide important insights to consider in planning for the next viral pandemic.

## 1. Introduction

The novel severe acute respiratory syndrome coronavirus 2 (SARS-CoV-2) virus, a RNA virus belonging to the Betacoronavirus genus of the Coronaviridae family, is the seventh coronavirus known to infect humans [1]. On 5 May 2023, WHO declared that Coronavirus disease 2019 (COVID-19) is no longer a Public Health Emergency of International Concern [2]. Public interest in COVID-19 has also waned. Although there are currently no SARS-CoV-2 variants meeting the criteria for variants of concern [3], there remains two variants of interest (Omicron BA.2.86 and KP.3) and one variant under monitoring (Omicron XEC), each with the potential to escalate in severity. During the pandemic, a dizzying array of molecular, serological, and antigen assays were developed to assist in the screening and diagnosis of COVID-19, each having a wide range of sensitivities, specificities and run times [4]. Yet, it is vital for the medical community to continue to stay abreast of the latest developments regarding the virus and related testing to better prepare for the next pandemic. In this review, we briefly recapitulate the changes to laboratory practice during the pandemic, history of assay evaluations performed both early and later during the pandemic, the documented antibody responses to both mRNA and inactivated virus vaccines with their real-world effectiveness, and some thoughts on vaccine efficacy and protective antibody levels.

## 2. Changes to Laboratory Work Practices During the Pandemic

Staffing and laboratory practices had to be modified extensively during the pandemic [5], with changes made to each part of the total laboratory testing process, including sample handling and collection. Thankfully, the major pre-analytical safety concerns of sample collection were addressed by World Health Organization guidelines early in the pandemic [6]. Vaccination was mandatory for all healthcare workers in our country (Singapore) in January 2021. Staff were segregated into separate working groups to reduce contact and prevent transmission of COVID-19. All active cases underwent home quarantine, with regular COVID-19 screening instituted across the lab using both PCR and rapid COVID-19 tests. In addition, staff also had to use personal protective equipment at all times. In addition, other aspects such as social distancing, limitations on the number of people using common facilities such as the pantry, and regular self-temperature logging had to be enforced.

Regular screening of staff using lateral flow immunoassays (LFIAs) became routine during the pandemic. Indeed, studies [7] demonstrate the effectiveness of frequent LFIA testing, showing that twice-weekly LFIA testing could diagnose infection even before the appearance of 70–80% of viable virus, as well as within 24–48 h after the first quantitative PCR positivity until 24–72 h cessation of viable virus. When our laboratory experienced an active case of COVID-19 in August 2021, staff (*n* = 161) were screened with LFIAs daily for 2 weeks, then twice weekly till November 2021, and then thrice weekly for 2 weeks in December 2021. Staff also underwent immediate RT-PCR testing on days 4, 7 and 10 after the active case detection. When we compared the outcome of twice and thrice weekly LFIA testing (SD Biosensor Standard Q and Abbott Pandbio COVID-19 Ag) to RT-PCR testing [8], both screening regimens demonstrated similar performance, with no cases detected with both regimens. Frequent LFIA testing thus represented a cost-effective approach to mass screening, and was essential to the prevention of workplace transmission between staff in healthcare institutions.

## 3. Assay Evaluations During the Pandemic

Viral RT-PCR testing was the gold standard test used to diagnose COVID-19. However, RT-PCR testing suffered from long turnaround times, high costs and limited reagent supply. First generation LFIA point-of-care tests (POCTs) COVID-19 antibody tests had poor performance up to April 2020 [9], and there was an urgent need for more sensitive rapid tests.

### 3.1. Early Antibody Testing in the Pandemic

Early in the pandemic, SARS-CoV-2 antibody responses were quickly shown to develop several days after symptom onset—within 17 to 19 days for IgG [10]. To provide earlier supportive findings than RT-PCR testing, we evaluated the early semi-quantitative Elecsys Anti-SARS-CoV-2 (Roche) electro-chemiluminescent antibody (total nucleocapsid) across 3 local institutions in early 2020 compared to RT-PCR [11]. Residual sera from other biochemical tests were tested in those with RT-PCR results. We used days post-first positive RT-PCR (POS) as a more objective milestone for disease onset, as it precluded confounding issues with asymptomatic/pre-symptomatic cases. The Elecsys Anti-SARS-CoV-2 (anti-N) assay initially had an excellent specificity (99.86%) and sensitivity of 97.1% at 14 days POS, improving to 100% by ≥21 days POS. However, within the first week POS, only 48.2% of patients had positive antibodies. In addition, we assessed the assay for cross-reactivity with several other viral infections. Out of 315 healthcare workers (HCWs) who had received their annual influenza vaccination 4 weeks prior to testing, none were reactive for anti-SARS-CoV-2, and only 1 out of 13 SARS-CoV-2 negative subjects positive for HBsAg had a false positive anti-SARS-CoV-2 result.

We followed-up with an evaluation of the (semi-quantitative) Abbott Architect SARS-CoV-2 IgG (anti-N) chemiluminescent microparticle immunoassay [12], which also showed excellent sensitivity of 96.7% (≥14 days POS). In both assays, antibody development was only evident after the first week POS.

However, the drawback of these early assays was that they assessed total antibodies (IgM and IgG) as well as nucleocapsid and spike antibodies. In addition, they were not sensitive enough to be used before 14 days POS. Another study [13] also showed that the median time to total antibody, IgM or IgG seroconversion was 11-, 12- and 14-days, with the antibody assays only having a positive rate of 38.3% within 7-days of symptom onset, improving to 90% by day 12 after symptom onset. Indeed, a Cochrane systematic review on SARS-CoV-2 antibody tests at that time also confirmed the low sensitivity of antibody tests in the first week after symptom onset but was useful 15 days after symptom onset [9], with the combination of IgG/IgM having a sensitivity of only 30.1% for the first week, and then improving to 91.4% at 15–21 days. Several studies on the early assays and early SARS-CoV-2 findings are summarized in Table 1 below. Given the uncertainty of the impact and virulence of COVID, health authorities had to vaccinate as many as possible. While antibody testing at that time may not have been useful for diagnosis of COVID-19, it could have been used to triage those for vaccination, if vaccines were in limited supply.

By early 2021, new POCTs for SARS-CoV-2 antibodies had been developed. We evaluated the Abbott Panbio COVID-19 IgG/IgM Rapid Tests and the Roche SARS-CoV-2 Rapid Antibody tests [14]. Both the Abbott Panbio COVID-19 IgG/IgM rapid test and the Roche SARS-CoV-2 Rapid Antibody test are lateral flow qualitative immunochromatographic assays. The performance of these tests versus the previous central laboratory tests [11,15] were found comparable only after 14 days POS (97.5–100% concordance). These newer POCTs still lacked sensitivity in the first week POS (only 78.5–80.0% concordance), with all assays having a sensitivity of <50% within the first week of infection, the Abbott POCT had only 10.8% sensitivity in the first week of infection, and the Roche POCT had a sensitivity of 18.5% in the first week.

**Table 1 vaccines-13-00009-t001:** Summary of early studies on SARS-CoV-2 antibody assays and COVID-19.

Study	Assays Used	Results
Zhao; et al. [13]	Total antibodies, IgM and IgG ELISA (Beijing Wantai Biologial Pharmacy Enterprise Co., Ltd. Beijing, China.)	➢ Total antibodies/IgM/IgG median seroconversion 11/12/14 days. ➢ Presence of antibodies was <40% within 1 week of onset, increasing to 100/94.3/79.8% by 15 days (total/IgM/IgG) ➢ Higher antibody titres were associated with a worse clinical classification (*p* = 0.006)
Deeks; et al. [9]	25 commercial tests and numerous in-house assays	➢ Pooled results for showed <30.1% sensitivity during the first week from symptom onset. ➢ 3rd week onwards: sensitivity 91.4/88.2/75.4/98.7/98.1% for IgG + IgM/IgG/IgM/IgA/total antibodies.
Favresse; et al. [15]	Elecsys anti-SARS-CoV-2 assay (Roche Diagnostics)	➢ Sensitivity improved to 91.7% ≥14 days after positive detection by RT-PCR. ➢ At <14 days, a lower cutoff improved sensitivity from 46.7/74.3% to 48.9/85.7%
Mizumoto; et al. [16]	RT-PCR testing	➢ 634/3711 individuals COVID-19 positive after 2 weeks quarantine. ➢ 328 cases were asymptomatic but RT-PCR positive.
Bryan; et al. [17]	Abbott Architect SARS-CoV-2 IgG assay (Anti-nucleocapsid)	➢ Sensitivity 53.1% at 7 days post symptom onset, improving to 96.9% at 14 days. ➢ 22 of 88 individuals positive anti-SARS-CoV-2 IgG on day 1 RT-PCR positive.
National SARS-CoV-2 Serology Assay Evaluation Group [18]	Abbott SARS-CoV-2 IgG, LIAISON SARS-CoV-2 S1/S2 IgG, Elecsys Anti-SARS-CoV-2, Siemens SARS-CoV-2 total assay, novel Oxford ELISA.	➢ Sensitivity 92.7–98.8% at ≥14 days post-symptom onset. ➢ Antibody responses reached a maximum by day 27 and sustained up to 82 days post positive RT-PCR. ➢ Discordance between assays was mostly seen in samples before 20 days post-symptom onset: 30% discordance, as opposed to 9% discordance after 20 days.
Whitman; et al. [19]	Epitope ELISA and In-house ELISA for IgM or IgG	➢ Sensitivity within the first 10 days for IgM/IgG: 39.3–80.6% (Epitope ELISA), and 35.7–72.2% (In-house ELISA) ➢ Sensitivity IgM only: 17.9–52.8% (Epitope ELISA)
Bastos; et al. [20]	ELISAs, LFIAs and CLIAs	➢ LFIA methods: IgG + IgM, pooled LFIA sensitivity is only 66%, CLIA was 97.8%, and ELISA 84.3% ➢ Sensitivities lower if using IgM by itself: ELISA 81.1%, LFIA 61.8%, CLIA 84.3%. ➢ Sensitivity lowest in the first week for all assays: ELISA 23.7%, LFIA 13.4%, and CLIA 53.2%.

Abbreviations: ELISA: enzyme-linked immunosorbent assay, RT-PCR: reverse transcription-polymerase chain reaction, COVID-19: Coronavirus disease 2019, SARS-CoV-2: Novel severe acute respiratory syndrome coronavirus 2, LFIA: Lateral flow immunoassay, CLIA: Chemiluminescent immunoassay.

### 3.2. IgM and IL6 in the Assessment of COVID-19

We had intended to utilize IgM positivity as a marker for early infection. However, IgM-bands on both POCTs displayed poor sensitivity even after 2 weeks POS (<50%). IgM assessment was still not useful in a subsequent evaluation using a more sensitive automated SARS-CoV-2 IgM assay [21], with 48.9% of cases not developing a detectable IgM even >14 days POS (sensitivity only 77.8%), and 16% (13/81) of positive cases having IgG positivity without IgM. Other studies reported SARS-CoV-2 IgG developing concomitantly or before IgM in the majority of cases [22], with IgM/IgG having a cumulative seroprevalence of 44/56% on day 7 after symptom onset. Consequently, as the pandemic progressed, the demand for IgM assays gradually declined as well.

Another analyte that received interest early in the pandemic was IL6, as early studies [23] suggested that elevated IL6 in patients with COVID-19 could predict survival outcomes and intensive care unit admission. Since tocilizumab (IL6 inhibitor) was a possible therapy for severe COVID-19 in April–June 2020, we evaluated the Roche IL6 chemiluminescent immunoassay [24]. When subsequent trials showed that IL6 inhibition with Tocilizumab was ineffective [25], interest in IL-6 assessment waned, although it remained a possible index for admission to intensive care in those with the highest IL-6 levels [24].

### 3.3. Antibody Assays in the Later Pandemic: The Spike Antibody Assays

The drawbacks of SARS-CoV-2 antibody assays up to this point were that they were all semi-quantitative assays targeting nucleocapsid antibodies, whereas SARS-CoV-2 spike proteins were responsible for viral ingress into host cells and the main target for neutralization [26,27,28]. mRNA vaccines targeting the spike protein were introduced by end-2020, and by mid-2021, we evaluated the next generation SARS-CoV-2 antibody assays that reported quantitative spike antibodies (S-Ab) (Roche Elecsys Anti-SARS-CoV-2 S assay) and neutralizing antibodies (N-Ab) (Snibe SARS-CoV-2 S-Receptor Binding Domain (RBD) IgG chemiluminescent immunoassay) [29]. Both assays had excellent specificity (Roche S-Ab 100%, Snibe N-Ab 92.0%), with the new Roche assay improving upon the previous generation, especially in the first 2 weeks post-disease onset: sensitivity of the Roche S-Ab was 48.1/79.5% in the 1st and 2nd weeks post-RT-PCR positivity, compared to nucleocapsid antibodies (43.3/74.4%). It was only after 2 weeks POS were the sensitivities of the two assays over 90%. Again, lower, optimized cutoff were required before sensitivity in early infection could be improved (e.g., with a revised cutoff 0.4 U/mL, the Roche S-Ab sensitivity improved to 56.7/82.1% in the first 2 weeks). The results of our analysis, together with the introduction of mandatory mRNA vaccination in Singapore from January 2021, supported our laboratory’s decision to offer S-Ab assays to monitor post-vaccination responses. Table 2 summarizes some of the studies comparing S-Ab and nucleocapsid antibody performance.

### 3.4. The Use of LFIA Antigen Tests in the Pandemic

LFIA antigen tests were introduced between 2020 and 2021. Several studies examined their use in public health screening (see Table 3). Their effectiveness in public health screening could not be underestimated. Within 1 week in Slovakia, mass testing (between 31 October to 1 November and 7 to 8 November) reduced the observed prevalence of COVID-19 by 58% [37]. As mass vaccination increased, the prevalence of COVID-19 decreased. However, screening with a single test in a low prevalence setting results in a low positive predictive value [38]. This in turn leads to more false positive results, with serious consequences for the health system, including the removal of patients from transplant lists, delayed surgeries, unnecessary delays in hospital discharge, and increased staff activity in nursing homes [39]. To improve false-positive rates, high specificity tests are preferable. Since the COVID-19 viremia peaks several days before to 5 days after symptom onset [40,41], the low sensitivity of initial tests can be compensated by a 2nd or 3rd test every 24 h after the onset of symptoms. Alternatively, a secondary confirmatory test can be performed in an orthogonal fashion for those initially antigen positive [38,42,43]. SARS-CoV-2 antigen tests, when used with molecular testing in an orthogonal fashion also achieves better positive predictive values (PPV) in low prevalence settings [38]. Even in settings with 0.1% disease prevalence, models of antigen testing (sensitivity 69.2%, specificity 99.1%) combined with molecular testing (sensitivity 88.1%, specificity 97.2%) could improve the PPV from 3.1% (molecular testing only) and 7.1% (antigen testing only) to 70.8% (antigen testing followed by molecular testing).

### 3.5. Automated SARS-CoV-2 Antigen Tests

More sensitive centralized laboratory SARS-CoV-2 antigen tests began to appear with a turnaround time of <1 h in late 2021 [52]. This allowed the greater usage of rapid and sensitive antigen testing especially for inpatient populations, either in place of the more expensive RT-PCR or to precede RT-PCR results which may take several hours or days. The Roche SARS-CoV-2 serum antigen assay [52] had good agreement with the RT-PCR (Spearman r = 0.90), with sensitivities of 42.6% and specificities of 99.7%. In those cases with high viremia (cycle threshold counts of <30), the sensitivity improved to 94.7%, and sensitivity was 62.5% within the first week of disease onset. The improved sensitivity of these antigen assays, in tandem with greater viremia was also observed in other studies (see Table 4). It is noteworthy that truly infectious patients have with greater threshold counts and would be easily detected by these rapid antigen tests. Besides, antigen testing was much cheaper than RT-PCR tests. Thus, it was more feasible to use serial antigen tests for first line screening with comparable or improved sensitivity [8]. Thus, for COVID patients admitted to hospital, serial lab-based 24×7 COVID-19 antigen testing could guide appropriate ward placement with a rapid turnaround time of under 1 h. This also applied to patients who became ill within the first week of disease.

## 4. Studies on Post-Vaccination SARS-CoV-2 Antibody Kinetics

### 4.1. Antibody Responses to mRNA Vaccines

As vaccinations became more prevalent, it was recognised that the primary response to the new mRNA vaccines was a sharp rise in spike IgG antibodies [57,58], cementing the test’s position as an important marker of vaccine response. As there was a paucity of information on antibody responses post-vaccination in Asian subjects, we studied the post-vaccination antibody kinetics in our local population. The older nucleocapsid antibody tests were used to identify subjects who were COVID-19 naïve. Using new S-Ab (Roche total S-Ab, Abbott IgG S-Ab and Abbott IgM S-Ab) and N-Ab (Snibe) assays [59], we evaluated antibody titres up to 90 days post-vaccination. Total S-Ab/IgG S-Ab/N-Ab peaked at 20 days post-vaccination, with a significant decline by day 40 post-vaccination (median 23.0% decrease for Roche total antibodies, and 34.4% for Abbott IgG) and plateauing by day 60–90, but remaining positive with a high titre (median 1069 BAU/mL for total antibodies, and 528 BAU/mL for IgG). On the other hand, IgM declines rapidly with a rising number of negative cases from day 20–90; only 9.4% (3/32) of subjects were still positive for IgM by day 90. Subjects < 50 years old had significantly higher total/IgG S-Ab and N-Abs than older subjects; both total S-Ab (r = 0.80) and IgG S-Ab (r = 0.85) had good agreement with N-Abs. The median total S-Ab, IgG S-Ab and N-Ab were all well above the assay positivity thresholds even at 180 days post-vaccination [60]. We estimated that the half-lives of total S-Ab/IgG S-Ab/N-Ab were 90/33/56 days, and by 180 days after complete vaccination with 2 doses of BNT162b2 mRNA vaccine, T-Ab/IgG/N-Ab were all still reactive (T-Ab 186 U/mL, IgG 617 AU/mL and N-Ab 0.39 ug/mL), but IgM was negative in all samples.

As the new Omicron variant gained prominence by late 2021, a third booster dose in vaccinated individuals was recommended [61], with evidence of efficacy against the new variant [62]. We also studied the spike antibody kinetics up to 90-days post-booster vaccination [63]. By 30 days post-booster, total S-Ab/N-Ab rose by 30/37x, again slowly decreasing by 90 days post-booster, but still remaining above the post-second dose peak, with estimated half-lives of 44 (S-Ab) and 58 (N-Ab) days. Even at 210 days post-booster [64], the total/IgG S-Ab and N-Ab titres remained significantly higher than pre-booster levels. Other studies [65,66] have shown that antibody responses remain robust, even up to 8–9 months after the 3rd vaccine dose. Despite this, several of the newer COVID-19 variants demonstrated the ability to “escape” vaccine-induced neutralizing antibody responses, with one study [65] showing that even after three doses of mRNA vaccine, 39.6 to 42.6% of vaccinees developed COVID-19 infections within 9 months of the last dose, and another study [66] showing that 51% of HCWs had breakthrough infections with the BA.1/BA.2 Omicron variants within 8 months of their last vaccination. In those who took a fourth dose of mRNA vaccine [67], a 9- to 10-fold increase in IgG against the RBD and neutralizing antibodies was seen. Notably, vaccine efficacy against any SARS-CoV-2 infection, including the Omicron variant, improved by 11–30% after a fourth booster dose. We provide a summary of several studies that also reported the evolution of post-vaccination antibody responses in Table 5 below.

### 4.2. Antibody Responses to Inactivated Virus Vaccines

Although mRNA vaccines were used to vaccinate and boost the majority of the population, inactivated virus vaccines were also commercially available. Some reviews [75] claimed that a two-dose schedule of CoronaVac had a 65.9% effectiveness in the prevention of COVID-19 in Chile (87.5% for hospitalization, 90.3% for ICU admission, and 86.3% for mortality). However, in Turkey, there was no significant difference in effectiveness for ICU admission/mortality between vaccinated and unvaccinated groups. Against the Delta variant, vaccine effectiveness against COVID-19 decreased from 74.5% at 1–2 months after the second dose, to only 30.4% 3–5 months after the second dose of CoronaVac, compared to a 90.8% to 79.3% drop in effectiveness in those vaccinated with BNT162b2 [76]. In Singapore, patients who received Sinovac-CoronaVac/Sinopharm were 2.4/1.6 times more likely to be infected with COVID-19, while those with Moderna vaccinations were 0.42 times less likely to contract infection [77].

We studied the antibody responses between 3 doses of mRNA vs. inactivated virus vaccine. The mRNA vaccine generated higher S-Ab/N-Ab responses than the inactivated virus vaccine at every time point up to 20 days post-booster [78], with mRNA vaccine recipients having 6.2/22.2/28.6-fold higher S-Ab and 2.5/9.2/22.2-fold higher N-Ab responses than inactivated virus vaccinees 10/20/20 days after their first/second/booster dose. Similarly, by 90 days after vaccine booster [79], S IgG was 1276 vs. 62.4 BAU/mL in mRNA vs. inactivated virus vaccinees, and N-Ab was 12.5 vs. 0.44 ug/mL. These findings were also supported in several other studies (see Table 6). This lends support for the choice of mRNA vaccines over inactivated virus vaccines for our doctors and patients.

### 4.3. The Assessment of Vaccine Effectiveness

As mass vaccination became widely implemented, vaccination programs undoubtedly saved many lives, with some estimates in the millions [83]. Although the practical benefits of vaccination were evident based on outcomes (see Table 7), it is difficult to come to a consensus on an analytical marker/cutoff for vaccine effectiveness. Some studies [84] indicate that antibodies directed against the S1 RBD account for 90% of serum neutralizing activity and are associated with decreased disease severity. Although promising as a marker for effectiveness/protection, it was never fully adopted in the face of the sheer heterogeneity of antibody responses to vaccination and infection. Nevertheless, one study [85] attempted to correlate neutralization levels to levels of protection, with a 20.2% of the mean convalescent level corresponding to a 50% protection rate. Another [86] found that a 67% protection against infection could be achieved with a IgG level of 94 BAU/mL (OmniPath 384 ELISA assay). Indeed, common spike antibody assays like the Roche/Abbott showed good agreement with SARS-CoV-2 neutralization assays (PPA 79.2/78.4% respectively) [87] and were commonly used as prognostic surrogates for vaccine effectiveness and protection. However, we have seen subjects with high antibody levels who subsequently contracted COVID-19.

## 5. Further Developments Post-Pandemic

Although public interest in COVID-19 has waned since the end of the pandemic, research into the SARS-CoV-2 virus and its effects continues. Although the broadly neutralizing antibodies that target the N-terminal/receptor-binding domains of the S1 subunit have been identified [93], there is difficulty in translating this information into the corresponding effectiveness. The specific neutralizing antibody/antibodies levels that indicate protective status for SARS-CoV-2 has yet to be ascertained. The SNIBE neutralizing antibody assay has also been discontinued because of low demand. Many of the neutralizing antibody assays are in-house tests [65], with different targets and performance. A study [94] comparing Roche SARS-CoV-2/Abbott IgG/DiaSorin SARS-CoV-2 IgG antibody assays showed that even with conversion to BAU/mL (referenced to the WHO standard), the results between analysers was not interchangeable (Abbott × Roche *p* = 0.89, and DiaSorin × Roche *p* = 0.87), with a value of 160/500 BAU/mL on Abbott/DiaSorin corresponding to 80 BAU/mL on the Roche assay before the second dose of vaccine. Furthermore, time-dependent changes in comparability were noted, with the Roche values lower than Abbott and DiaSorin three weeks after the first dose, to 5 times higher than Abbott 4 months after the third dose. Up till now, the degree and duration of immunity that antibodies confer, especially after vaccination, remains elusive. This is further confounded by every new round of COVID-19 variants, with breakthrough infections post-vaccination elevating antibody levels. For example, even with very high plaque reduction neutralisation test geometric mean titres after a booster dose with BNT162b2, the cumulative incidence with the new omicron BA.2 variant was roughly similar to subjects who had received a booster with coronavac instead (in subjects who previously received 2 doses of coronavac, incidence was 15.3/15.4% between those boosted with coronavac/BNT162b2) [95].

Research also continues in the development of new, more accessible virus assays for mass population screening. One study [96] examined the usage of Dimeric IgA tests in oral fluids for SARS-CoV-2 screening, which may be a more accessible sample for future viral infections. More studies would be needed before the relationship between IgA/IgG antibodies and different vaccines are fully elucidated. Another outstanding problem that limits test performance today is the lack of standardization and harmonization of serological tests [97,98], despite the FDA’s minimum performance criteria, WHO international standards and a serological sciences network (SeroNet) by the National Cancer Institute.

## 6. Conclusions

During the COVID-19 pandemic, laboratories faced unprecedented challenges, with major changes to work practices. In addition, the new assays that were rapidly introduced created a flurry of evaluations, but nonetheless allowed us to understand the antibody kinetics of SARS-CoV-2 infection and the responses to the newly developed mRNA vaccines. In general, S-Ab assays were eventually preferred to nucleocapsid antibody assays as the S-Abs had a robust response to vaccination, with mRNA vaccinees demonstrating superior antibody responses than inactivated virus vaccines. In addition, LFIAs also proved to be extremely effective in screening for COVID-19, especially when used serially over short time periods. All these developments had a profound impact on hospital admissions and rates of infection. We are thankful that we were able to extensively report these findings, to better prepare for future pandemics to come.

## Figures and Tables

**Table 2 vaccines-13-00009-t002:** Summary of studies comparing S-Ab and Nucleocapsid antibody assays.

Study	Assays Used	Results
Kening; et al. [30]	Magnetic chemiluminescence enzyme immunoassay (Shenzhen YHLO Biotech; Nanjing RealMind Biotech), iFlash3000 automated analyzer.	➢ There was a higher prevalence of total IgG antibodies than IgM in COVID-19 patients (IgG 70.83% vs. IgM 39.58%). ➢ On admission, patients with severe COVID-19 had lower S/RBD/N IgG than mild/moderate cases, with S IgG having the highest titres in both groups. ➢ RBD/S antibodies were higher than N during hospitalization and at discharge.
Lumley; et al. [31]	Abbott Architect SARS-CoV-2 IgG (Nucleocapsid), Spike IgG ELISA (Oxford)	➢ 6 months data of seroprevalence of 3276 UK HCWs for Nucleocapsid/Spike IgG. ➢ S IgG was stably detected in 94% of HCWs by 180 days. ➢ Nucleocapsid IgG peaked at 24 days after RT-PCR positive, then declined. N IgG half-life 85 days. ➢ Older age was associated with higher maximum Nucleocapsid IgG.
Lampasona; et al. [32]	Novel luciferase immunoprecipitation system, testing RBD/S1 + S2/Nucleocapsid antibodies.	➢ Observational cohort study of 509 COVID-19 subjects (including 139 diabetics), ➢ Positivity for IgG against the spike RBD was the most predictive of survival rate compared to Nucleocapsid IgG (lower HR for mortality of 0.4, S1 + S2 HR 0.53. N IgG was not associated with patient survival rates)
Burbelo; et al. [33]	Luciferase immunoprecipitation system assay, Berthold LB 960 Centro microplate luminometer	➢ 100 cross-sectional or longitudinal COVID-19 samples. ➢ At >14 days from symptom onset, Nucleocapsid antibodies had sensitivity 100%, S Abs only 91%.
Liu; et al. [34]	ELISA kits for N and S IgM/G (N Abs: Lizhu, Zhuhai, China, S Abs: Hotgen, Beijing China)	➢ Samples from 214 COVID-19 patients used to assess two ELISA tests. ➢ N vs. S IgG: 70.1% vs. 74.3% sensitivity ➢ N vs. S IgM: 68.2% vs. 77.1% sensitivity. ➢ N vs. S IgM and/or IgG: 80.4% vs. 82.2% sensitivity ➢ IgM sensitivities decreased after 35 days post-onset.
Elslande; et al. [35]	4 automated immunoassays: Roche (total Ig), Abbott (N IgG), Diasorin (S IgG), Snibe (N/S IgG) 2 ELISAs: Euroimmun (S1 and N), Mikrogen (N IgG)	➢ A comparison of antibody responses in 233 samples ➢ All assays had 100% sensitivity at 3 weeks after onset of symptoms ➢ Total IgG and nucleocapsid IgG assays had sensitivities of 25.6–32.6% in the first week, S IgG only 14.0–18.6% ➢ Overall, the total/nucleocapsid antibody assays had higher sensitivities (68.6–73.1%) than S IgG (63.2–64.6%)
Perkmann; et al. [36]	Roche total spike Ab, Abbott S IgG, DiaSorin TriS IgG, DiaSorin S1/2 IgG, Serion IgG.	➢ Sera from 69 COVID naïve individuals 21 +/− 1 days after their first dose of BNT162b2 vaccine. ➢ Despite conversion of all results to BAU/mL using recommended equations, assays were still not comparable, with mean values ranging from 66.3–1494.2 BAU/mL. ➢ Passing–Bablok regression confirmed relevant systemic proportional and constant differences between the 5 assays.

Abbreviations: COVID-19: Coronavirus disease 2019, S: Spike, RBD: Receptor binding domain, N: Nucleocapsid, SARS-CoV-2: Novel severe acute respiratory syndrome coronavirus 2, ELISA: enzyme-linked immunosorbent assay, HCW: Healthcare worker, HR: Hazard ratio.

**Table 3 vaccines-13-00009-t003:** Studies that examined the use of LFIA SARS-CoV-2 antigen tests.

Study	Assays Used	Results
Pekosz; et al. [44]	Quidel Lyra SARS-CoV-2 RT-PCR, BD Veritor rapid antigen test.	➢ Comparison of LFIA antigen/RT-PCR test results from 251 samples to viral cell culture. ➢ LFIA antigen tests had a greater PPV than RT-PCR: 90% vs. 73.7%. ➢ Antigen vs. PCR sensitivity: 96.4% vs. 100%. ➢ Overall percentage agreement with viral culture was higher for the antigen test: 98.4% vs. 96.0%, at <8 days after symptom onset.
Larremore; et al. [45]	Nil	➢ Modelling study of how repeated population screening with a lower sensitivity test would affect infectiousness ➢ Marked reductions in total infectiousness were observed by testing daily or every third day with >80% infectiousness removed, ➢ For populations, daily to every third day testing kept total infections to near 0%, with reductions in effective reproductive numbers of >80%.
Centers for Disease Control and Prevention [46]	Nil	➢ If symptomatic, an initial antigen test should be performed. If negative, perform serial antigen testing (24–48 h between tests) or nucleic acid amplification testing to determine if there is evidence for COVID-19. ➢ If asymptomatic, a negative antigen test can be used to rule out evidence of SARS-CoV-2 infection. ➢ If antigen positive, consider as evidence of SARS-CoV-2 infection.
Dinnes; et al. [47]	49 different commercial SARS-CoV-2 antigen LFIAs	➢ Sensitivity was higher in symptomatic (73%) than asymptomatic (54.7%) subjects. ➢ Average specificity was high for symptomatic (99.1%) and asymptomatic (99.7%) subjects. ➢ Variations in sensitivity were marked, with 34.3–91.3% in symptomatic vs. 28.6% to 77.8% in asymptomatic patients. ➢ At 5% prevalence in symptomatic subjects, PPV was 89%, and at 0.5% prevalence if asymptomatic, PPV would be 38–52%
Leli; et al. [48]	LumiraDx SARS-CoV-2 Ag test	➢ 792 patients, with a prevalence of 21%. ➢ LFIA showed a sensitivity of 68.7%, but if used in symptomatic patients, it increased to 81%. Specificity was 95.2% ➢ PPV was 79.2%, but increased to 96.9% if used in symptomatic patients
Cerutti; et al. [49]	STANDARD Q COVID-19 Ag (SD-Biosensor, RELAB, I)	➢ 185 Emergency department presentations (for diagnosis), and 145 travelers returning from high risk countries (for screening) ➢ In all patients, sensitivity was 70.6%, specificity was 100%, negative/positive predictive value was 87.4/100% ➢ If prevalence was 0.5%, PPV would still be 100% in both diagnostic and screening groups
Smith; et al. [50]	Quidel SARS Sofia antigen fluorescent immunoassay	➢ Antigen test daily sensitivity in 43 adults declined with the decline in presence of infectious virus. ➢ Prior to detectable viral shedding, PCR had sensitivity of 0.65, compared to antigen tests 0.375. Once viral cultures were positive, both had sensitivities of >0.8, with no significant differences between them. ➢ However, when used daily to every third day, the sensitivity of the antigen test at any time was very high at 1.0, similar to RT-PCR testing.
Chu; et al. [51]	QuickVue At-Home OTC COVID-19 Test, Quidel corporation	➢ 225 subjects. ➢ A single antigen test only had a sensitivity of 50% during the infectious period, peaking at 77% 4 days after illness, decreasing to 16% by 11 days after onset. ➢ Specificity was high at 97%. ➢ Sensitivity was higher for symptomatic cases 53% vs. asymptomatic 20%. ➢ Performing 2 antigen tests 2 days apart was more sensitive (85%) than 2 tests on consecutive days (81%) and a single test (77%).

Abbreviations: SARS-CoV-2: Novel severe acute respiratory syndrome coronavirus 2, RT-PCR: reverse transcription-polymerase chain reaction, LFIA: Lateral flow immunoassay, PPV: Positive predictive value, Ag: Antigen, COVID-19: Coronavirus disease 2019.

**Table 4 vaccines-13-00009-t004:** Studies of central laboratory SARS-CoV-2 antigen tests.

Study	Assays Used	Results
Saito; et al. [53]	HISCL SARS-CoV-2 Antigen ELISA kit.	➢ 115 NP swabs from 46 COVID-19 patients with 69 controls. ➢ Antigen assay sensitivity of 95.4% in samples with higher copy numbers > 100 ➢ The sensitivity decreased to 16.6% if the copy number was <99. ➢ Specificity was excellent at 100% ➢ At copy numbers of <50, 81.8% of samples were negative.
Mitchell; et al. [54]	Quidel Sofia SARS IFA (Sofia)	➢ 148/144 symptomatic/asymptomatic subjects ➢ Antigen test sensitivity 87.8% in symptomatic patients, but 33.3% in asymptomatic. ➢ Sensitivity decreased > 5 days post symptom onset (90% before 5 days, 82% after) ➢ Specificity remained excellent at 100%.
Osterman; et al. [55]	Lumipulse G SARS-CoV-2 Ag (Fujirebio Inc), LIAISON SARS-CoV-2 Ag (Diasorin), Elecsys SARS-CoV-2 Antigen (Roche Diagnostics), SARS-CoV-2 Antigen ELISA (Euroimm)	➢ Comparison of four automated platforms for SARS-CoV-2 antigen tests ➢ Specificities of all automated Ag tests were high: 97.0–99.7% ➢ Sensitivity of all automated tests was low: 17.76 to 52.34%, which could be improved by optimizing manufacturer cut-offs (27.10–79.44%)
Fourati; et al. [56]	VITROS SARS-CoV-2 Antigen EIA test.	➢ Using Ct < 30 as a threshold, sensitivity was 97.0% in the retrospective cohort, 98.8% in the prospective. ➢ Using a Ct < 35 as a threshold, sensitivity was 82.8% in the retrospective cohort, 93.5% in the prospective. ➢ Sensitivity was greatest at days 4–7 (77.3%), vs. days 8–11/>12 (60.0/50.0%) in the retrospective cohort. ➢ Overall specificity remained at 100%. ➢ The assay could detect 90.4% of B.1.1.7/alpha, and 100% of B.1.351/beta

Abbreviations: SARS-CoV-2: Novel severe acute respiratory syndrome coronavirus 2, ELISA: enzyme-linked immunosorbent assay, NP: Nasopharyngeal, COVID-19: Coronavirus disease 2019, Ag: Antigen, Ct: Cycle threshold count.

**Table 5 vaccines-13-00009-t005:** Studies examining the course of antibody response to vaccination.

Study	Assays Used	Results
Doria-Rose; et al. [68]	RBD ELISA, pseudovirus neutralization assay, live-virus neutralization assay	➢ mRNA1273-elicited binding and neutralizing antibodies in a phase 1 trial, up to 180 days after the second dose. ➢ Geometric mean end-point titer for spike RBD antibodies remained high at day 209 (GMT 92,451, 18–55 y/o; GMT49,373, >71 y/o). ➢ The estimated half-life of RBD Abs after day 43 was 52 days.
Favresse; et al. [69]	Elecsys anti-SARS-CoV-2 NCP total qualitative ECLIA (nucleocapsid antibodies), Elecsys anti-SARS-CoV-2 spike total quantitative ECLIA.	➢ Multicentre examination of antibody responses in HCWs who had received 2 doses of BNT162b2 vaccine, up to 90 days. ➢ Maximum concentration of spike antibodies was 36 +/− 3 days after the first dose, with an estimated half-life of 55 days, in initially seronegative patients ➢ If initially seropositive, maximum antibodies reached at 24 +/− 4 days, half-life of antibodies also increases to 80 days. ➢ At 3 months, mean antibody decrease of 37.9% and 44.7% in seronegative and seropositive subjects. ➢ Nucleocapsid antibodies remained stable in seropositives, and persistently negative in seronegatives.
Levin; et al. [70]	Access SARS-CoV-2 IgG (Beckman Coulter), Platelia SARS-CoV-2 Total Ab Assay (Bio-Rad)	➢ 6-month longitudinal prospective study in vaccinated HCWs (BNT162b2 vaccine, 2 doses) ➢ Antibodies peaked 4–30 days after the second dose, IgG then declined consistently over 6 months. N Abs declined rapidly in 3 months. ➢ IgG was still correlated with N Abs (Spearman rank 0.68–0.75) ➢ N Abs were lower in >65 y/o by the end of 6 months.
Bayart; et al. [71]	Elecsys SARS-CoV-2 total antibodies (Roche), Architect SARS-CoV-2 IgG (Abbott), pseudo-virus neutralization test	➢ An observation of the total/IgG and Nab levels in 231 HCWs after 2 doses of BNT162b2 vaccine. ➢ By day 180, total antibodies had decreased by 55.4% (seronegative) and 74.8% (seropositive). ➢ IgG had decreased by 89.6% (seronegative) and 79.4% (seropositive) ➢ IgG half life was estimated to be 21 days in seronegative, and 53 days in seropositive. ➢ 45% of subjects were negative for Nabs by 180 days.
Eliakim-Raz; et al. [72]	SARS-CoV-2 IgG II Quant assay (Abbott)	➢ An observational study of S IgG titres before and after a third dose of BNT162b2. ➢ Median titers of antibodies increased from 440 AU/mL to 25,468 AU/mL.
Lin; et al. [73]	Abbott SARS-CoV-2 IgG II Quant assay, Abbott Architect.	➢ An observational study of 399 participants: prime vaccination with ChAdOx1-S/mRNA-1273, booster with ChAdOx1-S/mRNA-1273 in different combinations, 18 of whom were on immunosuppressant therapy ➢ Lower anti-SARS-CoV-2 IgG S-Ab were found in immunosuppressed patients before booster (36.39 vs. 83.84 BAU/mL) ➢ 4 weeks post-booster, although the titers were numerically lower in immunosuppressed patients compared to controls (1590.61 vs. 1918.38 BAU/mL), immunogenicity nonetheless did improve to rise above pre-booster levels.
Grob; et al. [62]	SARS-CoV-2 ELISA (Euroimmun), SARS-CoV-2 (Delta) surrogate virus neutralization test (GenScript)	➢ Prospective study investigating humoral/cellular immune responses to a heterologous vaccination regimen (ChAdOx1-S prime, then BNT162b2) ➢ Strong neutralization titres were noted 2 weeks after booster, IgG was detectable in all participants within 6–11 days (increased 134-fold) ➢ Over 17 weeks, titers decreased over time, but even at the end point, all analysed sera still retained neutralizing activity against Delta variants.
Lee; et al. [74]	Elecsys Anti-SARS-CoV-2 S assay (Roche)	➢ Quantitative analysis of S-Ab after 2 doses of ChAdOx1-S with a BNT162b2 booster in 149 HCWs ➢ Median S-Ab had decreased 4 months after the third dose (1st month 17,777 U/mL, decrease to 3673 U/mL) in uninfected individuals. ➢ 58/149 subjects had breakthrough infections during the 4-month period, and their S-Ab was higher at 4 months than the non-infected group (19.539 vs. 3673 U/mL)

Abbreviations: RBD: Receptor binding domain, ELISA: enzyme-linked immunosorbent assay, GMT: Geometric mean titre, Abs: Antibodies, SARS-CoV-2: Novel severe acute respiratory syndrome coronavirus 2, ECLIA: Electrochemiluminescent immunoassay, HCW: Healthcare worker, N: Neutralizing, S: Spike.

**Table 6 vaccines-13-00009-t006:** Studies comparing the antibody response between inactivated and mRNA vaccines.

Study	Assays Used	Results
Lim; et al. [80]	Spike RBD antibody ELISA, surrogate virus neutralization assay, plaque reduction neutralisation test.	➢ Immune responses in 93 HCWs who received either the BNT162b2 (*n* = 63) or Coronavac vaccines. ➢ After the second dose of mRNA vaccine, geometric mean PRNT50 titre was 269, and geometric mean PRNT90 titre was 113. ➢ Geometric mean PRNT50 titre was 27 after a second dose of inactivated virus vaccine, and geometric mean PRNT90 was only 8.4. ➢ RBD ELISA (in OD450 values): mRNA vaccines had a value of >3 after the second dose, Coronavac only < 2.
Barin; et al. [81]	VIDAS SARS-CoV-2 IgG	➢ A prospective comparison of immunogenicity between 2 doses of BNT162b2 (*n* = 106), ChAdOx1-S (*n* = 56), and Coronavac (*n* = 222) vaccines, and a booster dose of BNT162b2/Coronavac after 2 doses of Coronavac ➢ BNT162b2 induced the highest antibody response: 35.3 AU at 1 month, 19.2 AU at 3 months ➢ ChAdOx1-S also had a modest antibody response: 17.1 AU at 1 month, 6.5 AU at 3 months ➢ Coronavac had the lowest antibody response: 11.3 AU at 1 month, and 2.4 AU at 3 months. ➢ Coronavac had the fastest decline in antibodies ➢ 3 months after the second dose, only 60% of older patients (>60 y/o, *n* = 127) were still seropositive. ➢ Boosting with BNT162b2 after 2 doses of Coronavac significantly improved antibody levels, with a 7.9-fold increase in titres.
Khong; et al. [82]	Live virus microneutralization assay	➢ Investigation of a heterologous prime-boost strategies to enhance immune responses in 37 patients. ➢ Boosting with a BNT162b2 vaccine after 2 doses of Coronavac elicited a higher immunogenicity against Wild type, beta and Delta variants, with some protection against Omicron. ➢ Boosting with mRNA vaccine: 32-fold GMT increase against delta variant, 4.76-fold GMT increase against Omicron ➢ Boosting with Coronavac: 2.52-fold GMT increase against delta, 1.17-fold GMT increase against Omicron.

Abbreviations: RBD: Receptor binding domain, ELISA: enzyme-linked immunosorbent assay, HCW: Healthcare worker, PRNT: Plaque reduction neutralization titre, GMT: Geometric mean titre.

**Table 7 vaccines-13-00009-t007:** Outcomes of mass vaccination.

Study	Vaccine Used	Results
Bernal; et al. [88]	2 doses: BNT162b2 or ChAdOx1-S	➢ A test negative case-control study observing community testing for COVID-19 in 156,930 adults aged > 70 years old. ➢ 1 dose BNT162b2: effectiveness reached a maximum of a 70% (reduction of OR from 1.48 to 0.41) at preventing COVID-19. ➢ From 14 days after the second dose, effectiveness was 89% (OR 0.15), BNT162b2: 61% effectiveness 28–34 days later, ChAdOx1-S 60%. ➢ BNT162b2: 43% reduced risk of hospital admission, ChAdOx1-S 37%, BNT162b2: 51% reduced risk of death.
Thompson; et al. [89]	1 (12.1%) or 2 (62.8%) doses: BNT162b2 or mRNA-1273.	➢ Prospective cohort study of 3950 HCWs ➢ Unvaccinated: 1.38 infections/1000 person-days, vs. 0.04 infections/1000 person-days > 14 days after second dose (0.19 infections/1000 person days for only 1 dose).
Haas; et al. [90]	2 doses of BNT162b2	➢ 95.3% effectiveness 7 days or longer after the second dose: infection incidence rate of 3.1 vs. 91.5 per 100,000 person-days in vaccinated vs. unvaccinated. ➢ 97.0% effectiveness against symptomatic infection, 91.5% against asymptomatic infection. ➢ 97.2% (4.6 vs. 0.3 per 100,000 person days) against hospitalization, 97.5% against severe/critical admission ➢ 96.7% (0.6 vs. 0.1 per 100,000 person days) for COVID-19 related death.
Bar-On; et al. [91]	3 doses of BNT162b2	➢ Rates of infection were lower in the booster group than the non-booster group by a factor of 11.3 ➢ Severe illness was reduced by a factor of 19.5.
Mohammed; et al. [92]	Nil	➢ mRNA vaccines had very high efficacy at preventing infections: Pfizer 95.0% at >7 days after vaccination, Moderna 94.1% at >14 days after vaccination ➢ mRNA vaccines had superior effectiveness compared to inactivated vaccines in preventing infection: 91.3% compared to 73.8%. ➢ Pfizer vaccines had a 75.6–100% effectiveness at reducing the number of hospitalizations, 74–100% for mortality.

Abbreviations: COVID-19: Coronavirus disease 2019, OR: Odds ratio, HCW: Healthcare workers.

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
