# Peer review of "Reflections on COVID-19: A Literature Review of SARS-CoV-2 Testing"

_vaccines, 2024, doi:10.3390/vaccines13010009_

Round 1
Reviewer 1 Report
Comments and Suggestions for Authors
This review provides a narrative overview of the evolution and application of SARS-CoV-2 commercial immunoassays, focusing on the early pandemic to the vaccine era, particularly mRNA vaccines. The manuscript maintains a well-balanced scope but is not deep in detail.
I suggest revising the terminology of the commercial tests by using the correct name.
Comments.
1. Incorrect statement about the pandemic status (Line 36):
The statement "The World Health Organization declared that the Coronavirus Disease 2019 (COVID-19) pandemic over in May 2023" is incorrect.
On 5 May 2023, WHO declared that COVID-19 is no longer a Public Health Emergency of International Concern (PHEIC), but the pandemic status has not been officially revoked.
Reference: https://www.who.int/europe/news-room/05-05-2023-statement-on-the-fifteenth-meeting-of-the-international-health-regulations-(2005)-emergency-committee-regarding-the-coronavirus-disease-(covid-19)-pandemic
2. Line 84, Roche immunologic assessment tools:
Replace "The Roche total antibody" with the specific test name: Elecsys Anti-SARS-CoV-2 (anti-N) or Elecsys Anti-SARS-CoV-2 S (anti-RBD).
This clarification avoids confusion, as Roche manufactures multiple tests targeting different SARS-CoV-2 proteins.
3. Lines 92–93, Abbott immunologic assessment tool.
Confirm whether this refers to the Abbott Architect SARS-CoV-2 IgG (anti-N) test.
Specify the exact target (e.g., anti-N, anti-S, anti-RBD) and technology platform of Chemiluminescent Microparticle Immunoassay (CMIA) instead of using the generic term "chemiluminescent."
Reference: https://www.fda.gov/media/137383/download
5. Line 148, "... quantitative spike antibodies (S-Ab) (Roche S-Ab assay)"
Was this test named Elecsys Anti-SARS-CoV-2 S (Roche)?
References: https://diagnostics.roche.com/global/en/products/lab/elecsys-anti-sars-cov-2-s-cps-000616.html
https://www.fda.gov/media/144037/download
6. Table 5, SARS-CoV-2 protein used in sVNT.
Specify the SARS-CoV-2 protein variant used in the GenScript surrogate virus neutralisation test (e.g., wild-type, Delta, Omicron BA.1).
7. (Optional) I think you may add more sections about humoral immunity in long-term follow-up (both natural infection, vaccination, and hybrid immunity) which could gain more interesting content and attract more readers.
Terminology recommendations.
1. Line 80, "Roche anti-SARS-CoV-2"
Use the specific test name: Elecsys Anti‑SARS‑CoV‑2 (Roche).
References:
https://diagnostics.roche.com/global/en/products/lab/elecsys-anti-sars-cov-2-cps-000273.html
https://www.fda.gov/media/137605/download
2. ChAdOx1-S
Use ChAdOx1-S instead of ChAdOx1 to specify the version containing the SARS-CoV-2 spike protein gene.
Rationale: ChAdOx1 refers to the adenoviral vector platform, while ChAdOx1-S specifies the spike protein encoding vector.
Author Response
Comments from Reviewer 1
- Incorrect statement about the pandemic status (Line 36): The statement "The World Health Organization declared that the Coronavirus Disease 2019 (COVID-19) pandemic over in May 2023" is incorrect.
On 5 May 2023, WHO declared that COVID-19 is no longer a Public Health Emergency of International Concern (PHEIC), but the pandemic status has not been officially revoked.
Reference: https://www.who.int/europe/news-room/05-05-2023-statement-on-the-fifteenth-meeting-of-the-international-health-regulations-(2005)-emergency-committee-regarding-the-coronavirus-disease-(covid-19)-pandemic
Reply: We have corrected the above statement to the suggested correction and added the reference.
- Line 84, Roche immunologic assessment tools: Replace "The Roche total antibody" with the specific test name: Elecsys Anti-SARS-CoV-2 (anti-N) or Elecsys Anti-SARS-CoV-2 S (anti-RBD).
This clarification avoids confusion, as Roche manufactures multiple tests targeting different SARS-CoV-2 proteins.
Reply: We have corrected the terminology accordingly to Elecsys Anti-SARS-CoV-2 (anti-N).
- Lines 92–93, Abbott immunologic assessment tool. Confirm whether this refers to the Abbott Architect SARS-CoV-2 IgG (anti-N) test. Specify the exact target (e.g., anti-N, anti-S, anti-RBD) and technology platform of Chemiluminescent Microparticle Immunoassay (CMIA) instead of using the generic term "chemiluminescent."
Reference: https://www.fda.gov/media/137383/download
Reply: We have made the correction to Abbott Architect SARS-CoV-2 IgG (anti-N) chemiluminescent microparticle immunoassay.
- Line 148, "... quantitative spike antibodies (S-Ab) (Roche S-Ab assay)" Was this test named Elecsys Anti-SARS-CoV-2 S (Roche)?
References: https://diagnostics.roche.com/global/en/products/lab/elecsys-anti-sars-cov-2-s-cps-000616.html
https://www.fda.gov/media/144037/download
Reply: We have corrected this to the Roche Elecsys Anti-SARS-CoV-2 S assay.
- Table 5, SARS-CoV-2 protein used in sVNT. Specify the SARS-CoV-2 protein variant used in the GenScript surrogate virus neutralisation test (e.g., wild-type, Delta, Omicron BA.1).
Reply: Done.
- (Optional) I think you may add more sections about humoral immunity in long-term follow-up (both natural infection, vaccination, and hybrid immunity) which could gain more interesting content and attract more readers.
Reply: We agree with your recommendation. However, the article as it stands is quite lengthy at >7000 words. We intend to follow up on this in a future publication.
Terminology recommendations.
- Line 80, "Roche anti-SARS-CoV-2": Use the specific test name: Elecsys Anti‑SARS‑CoV‑2 (Roche).
References: https://diagnostics.roche.com/global/en/products/lab/elecsys-anti-sars-cov-2-cps-000273.html
https://www.fda.gov/media/137605/download
Reply: We have changed the terminology accordingly.
- ChAdOx1-S: Use ChAdOx1-S instead of ChAdOx1 to specify the version containing the SARS-CoV-2 spike protein gene.
Rationale: ChAdOx1 refers to the adenoviral vector platform, while ChAdOx1-S specifies the spike protein encoding vector.
Reply: We have changed the terminology accordingly.
Reviewer 2 Report
Comments and Suggestions for Authors
In the manuscript by Lau et al., the authors reviewed the COVID-19 test assays, especially used in Singapore, including nucleocapsid/spike antibodies and automated antigen tests, from early pandemic to post-pandemic. During the evaluation, viral RT-PCR testing was used as the gold standard. Early antibody testing was focused on total SARS-CoV-2 antibodies, and both Roche assay and Abbott assays showed antibody development was only evident after the first week POS. They were not sensitive enough to be used before 14 days POS. The authors then showed later pandemic antibody assays with a focus on the spike antibody. The authors also included automated SARS-CoV-2 antigen tests. After that, the authors reviewed studies on pos-vaccination SARS-CoV-2 antibody kinetics and found mRNA vaccines better than inactivated virus vaccines. At last, the authors discussed the test assay developments post-pandemic. Some developments have been discontinued because of low demand, including some neutralizing antibody assays.
Generate speaking, it is a nice summary of the COVID-19 pandemic assay evaluation and may be helpful for future pandemic preparation.
Author Response
Comments from Reviewer 2
In the manuscript by Lau et al., the authors reviewed the COVID-19 test assays, especially used in Singapore, including nucleocapsid/spike antibodies and automated antigen tests, from early pandemic to post-pandemic. During the evaluation, viral RT-PCR testing was used as the gold standard. Early antibody testing was focused on total SARS-CoV-2 antibodies, and both Roche assay and Abbott assays showed antibody development was only evident after the first week POS. They were not sensitive enough to be used before 14 days POS. The authors then showed later pandemic antibody assays with a focus on the spike antibody. The authors also included automated SARS-CoV-2 antigen tests. After that, the authors reviewed studies on pos-vaccination SARS-CoV-2 antibody kinetics and found mRNA vaccines better than inactivated virus vaccines. At last, the authors discussed the test assay developments post-pandemic. Some developments have been discontinued because of low demand, including some neutralizing antibody assays.
Generally speaking, it is a nice summary of the COVID-19 pandemic assay evaluation and may be helpful for future pandemic preparation.
Reply: We thank you for your time and effort in reviewing our paper. Your encouragement is much appreciated.
Reviewer 3 Report
Comments and Suggestions for Authors
In the manuscript entitled “Reflections on COVID-19 testing”, Lau et al. provided a comprehensive literature review of SARS-CoV-2 testing. The authors took a systematic approach to examine the evolution of both PCR and antibody testing and how such testing played crucial roles in public policies and vaccine development. Overall, the manuscript is well written. There are a few typos in the manuscript. There is no reason to capitalize “Novel” in line 38 and “Elevated” in line 135. There are a total of 93 references cited in the text but only 92 are listed under References. The list of references appears to have become incorrect from #69 and on (#69 does not exist in the text). Additional proofreading would be beneficial. In addition, the disclosures are all missing (e.g. author contributions, funding, COI statement, etc). A considerable number of self-citations (n = 16) are also observed.
Author Response
Comments from Reviewer 3
In the manuscript entitled “Reflections on COVID-19 testing”, Lau et al. provided a comprehensive literature review of SARS-CoV-2 testing. The authors took a systematic approach to examine the evolution of both PCR and antibody testing and how such testing played crucial roles in public policies and vaccine development. Overall, the manuscript is well written. There are a few typos in the manuscript. There is no reason to capitalize “Novel” in line 38 and “Elevated” in line 135. There are a total of 93 references cited in the text but only 92 are listed under References. The list of references appears to have become incorrect from #69 and on (#69 does not exist in the text). Additional proofreading would be beneficial. In addition, the disclosures are all missing (e.g. author contributions, funding, COI statement, etc). A considerable number of self-citations (n = 16) are also observed.
Reply: We thank you for your helpful pointers. We have corrected the typos, added references for [73] and [98], and added the disclosures in the back matter.
Reviewer 4 Report
Comments and Suggestions for Authors
- The title should be more descriptive.
- Very short introduction without any references!
- Line 51, in our country, which country! Besides, the term “our” was repeated all over the manuscript.
- All the abbreviations should be written as full at the first mention (in the abstract and the manuscript).
- The paragraph (lines 58-70) should be re-written in a correct way.
- Tables (1-7) should be written in a simple form. The references in numbers, concise results, and the abbreviations under it.
- The conclusion should be re-written to summarize the topics of the article and give some recommendations.
- Author Contributions, Funding, Institutional Review Board Statement, Informed Consent Statement, Data Availability Statement, Acknowledgments, Conflicts of Interest should be completed.
Best wishes
Author Response
Comments from Reviewer 4
1: The title should be more descriptive.
Reply: We have enhanced the title of the article.
2: Very short introduction without any references!
Reply: We have extended the introduction, and included more references.
Line 51, in our country, which country! Besides, the term “our” was repeated all over the manuscript.
Reply: We have specified that “our country” refers to Singapore, and tried to reduce the use of “our” in the article as much as possible.
All the abbreviations should be written as full at the first mention (in the abstract and the manuscript).
Reply: We have edited accordingly.
The paragraph (lines 58-70) should be re-written in a correct way.
Reply: We have edited the paragraph.
Tables (1-7) should be written in a simple form. The references in numbers, concise results, and the abbreviations under it.
Reply: We have edited the tables accordingly.
The conclusion should be re-written to summarize the topics of the article and give some recommendations.
Reply: We have re-written the conclusion.
Author Contributions, Funding, Institutional Review Board Statement, Informed Consent Statement, Data Availability Statement, Acknowledgments, Conflicts of Interest should be completed.
Reply: We have added the back matter.
Round 2
Reviewer 4 Report
Comments and Suggestions for Authors
Thanks for your response.
The only thing that should be taken in consideration during the final revision is the method of writing the authors names of the references in the table (Only the first name of the author then et al without abbreviations).
Best wishes